# Deciphering the Impact of Early-Life Exposures to Highly Variable Environmental Factors on Foetal and Child Health: Design of SEPAGES Couple-Child Cohort

**DOI:** 10.3390/ijerph16203888

**Published:** 2019-10-14

**Authors:** Sarah Lyon-Caen, Valérie Siroux, Johanna Lepeule, Philippe Lorimier, Pierre Hainaut, Pascal Mossuz, Joane Quentin, Karine Supernant, David Meary, Laurence Chaperot, Sam Bayat, Flemming Cassee, Sarah Valentino, Anne Couturier-Tarrade, Delphine Rousseau-Ralliard, Pascale Chavatte-Palmer, Claire Philippat, Isabelle Pin, Rémy Slama

**Affiliations:** 1Inserm, CNRS, Team of Environmental Epidemiology Applied to Reproduction and Respiratory Health, IAB (Institute for Advanced Biosciences) Joint Research Center, University Grenoble Alpes, 38700 Grenoble, France; sarah.lyon-caen@univ-grenoble-alpes.fr (S.L.-C.); valerie.siroux@univ-grenoble-alpes.fr (V.S.); johanna.lepeule@univ-grenoble-alpes.fr (J.L.); jquentin@chu-grenoble.fr (J.Q.); karine.supernant@univ-grenoble-alpes.fr (K.S.); claire.philippat@inserm.fr (C.P.); ipin@chu-grenoble.fr (I.P.); 2Biological Ressources Centre (CRB), Grenoble University Hospital, 38700 La Tronche, France; plorimier@chu-grenoble.fr (P.L.); pmossuz@chu-grenoble.fr (P.M.); 3Inserm, CNRS, Team of Tumor Molecular Pathology and Biomarkers, IAB (Institute for Advanced Biosciences) Joint Research Center, University Grenoble Alpes, 38700 Grenoble, France; pierre.hainaut@univ-grenoble-alpes.fr; 4Pediatric Department, Grenoble University Hospital, 38700 La Tronche, France; sbayat@chu-grenoble.fr; 5CNRS, LPNC UMR 5105, University Grenoble Alpes, 38000 Grenoble, France; david.meary@univ-grenoble-alpes.fr; 6Inserm, CNRS, Team of Immunobiology and Immunotherapy in Chronic Diseases, IAB (Institute for Advanced Biosciences) Joint Research Center, University Grenoble Alpes, 38700 Grenoble, France; laurence.chaperot@efs.sante.fr; 7Etablissement Français du Sang Auvergne-Rhône-Alpes, Research and Development Laboratory, 38700 Grenoble, France; 8Inserm UA7, Synchrotron Radiation for Biomedicine Laboratory (STROBE), University Grenoble Alpes, 38000 Grenoble, France; 9National Institute for Public Health and the Environment, 3720 Bilthoven, The Netherlands; flemming.cassee@rivm.nl; 10Institute of Risk Assessment Studies, Utrecht University, 3508 Utrecht, The Netherlands; 11UMR BDR, INRA, ENVA, Université Paris Saclay, 78350 Jouy-en-Josas, France; valentino.sarah@orange.fr (S.V.); anne.couturier-tarrade@inra.fr (A.C.-T.); delphine.rousseau@inra.fr (D.R.-R.); pascale.chavatte-palmer@inra.fr (P.C.-P.)

**Keywords:** DOHaD, environmental epidemiology, birth cohort, endocrine disruptors, atmospheric pollutants, exposome, child health

## Abstract

In humans, studies based on Developmental Origins of Health and Disease (DOHaD) concept and targeting short half-lived chemicals, including many endocrine disruptors, generally assessed exposures from spot biospecimens. Effects of early-life exposure to atmospheric pollutants were reported, based on outdoor air pollution levels. For both exposure families, exposure misclassification is expected from these designs: for non-persistent chemicals, because a spot biospecimen is unlikely to capture exposure over windows longer than a few days; for air pollutants, because indoor levels are ignored. We developed a couple-child cohort relying on deep phenotyping and extended personal exposure assessment aiming to better characterize the effects of components of the exposome, including air pollutants and non-persistent endocrine disruptors, on child health and development. Pregnant women were included in SEPAGES couple-child cohort (Grenoble area) from 2014 to 2017. Maternal and children exposure to air pollutants was repeatedly assessed by personal monitors. DNA, RNA, serum, plasma, placenta, cord blood, meconium, child and mother stools, living cells, milk, hair and repeated urine samples were collected. A total of 484 pregnant women were recruited, with excellent compliance to the repeated urine sampling protocol (median, 43 urine samples per woman during pregnancy). The main health outcomes are child respiratory health using early objective measures, growth and neurodevelopment. Compared to former studies, the accuracy of assessment of non-persistent exposures is expected to be strongly improved in this new type of birth cohort tailored for the exposome concept, with deep phenotyping and extended exposure characterization. By targeting weaknesses in exposure assessment of the current approaches of cohorts on effects of early life environmental exposures with strong temporal variations, and relying on a rich biobank to provide insight on the underlying biological pathways whereby exposures affect health, this design is expected to provide deeper understanding of the interplay between the Exposome and child development and health.

## 1. Introduction

### 1.1. Health Effects of Early-Life Environmental Exposures

The Developmental Origins of Health and Disease (DOHaD) concept states that the risk of chronic health disorders in childhood and adulthood, as well as possibly in the following generation(s), may be increased by environmental stressors during the period of development, which encompasses intra-uterine life and the first years of life [1,2,3]. Illustrations of the DOHaD concept in animal models exist for various types of environmental stressors, such as maternal diet during pregnancy [4,5], and chemicals. Regarding chemical factors specifically, a large number of toxicological studies showed biological or adverse health effects due to developmental exposures [6,7]. These toxicological studies relate to persistent chemicals such as polychlorinated biphenyls (PCBs) [6,7] and non-persistent chemicals, which include phthalates [8], phenols such as bisphenol A [9], parabens or triclosan [10], and many other chemical families. There is a family of environmental factors for which the short and possibly long-term effects of pregnancy exposure have been relatively little considered in toxicology, namely atmospheric pollutants. A few toxicological experiments have reported possible effects of pregnancy exposure to particulate matter, notably in mice [11,12]; additional studies are warranted here, in particular on animal models with a placenta closer to that of humans than the mouse placenta.

In humans, well-documented examples of associations between exposure to environmental factors during the developmental period and health in childhood and adulthood include studies on exposure to tobacco smoke [13,14], the drug diethylstilbestrol (DES) [15,16], the insecticide DDT [17], PCBs [18,19], metals such as lead [20] or mercury [21,22], perfluorinated compounds [23] or fine particulate matter (PM) exposure [24]. Schematically, there is clear human evidence of health effects for many of the persistent compounds identified as hazardous by toxicology. For the non-persistent chemicals, although policy measures can be taken without confirmation in humans of effects observed in animal experiments, from a scientific perspective, there is a need for accurate human studies. This is all the more true since the huge majority of currently marketed compounds in industrialized countries are non-persistent in the body. These raise specific issues in terms of exposure assessment in epidemiological studies (see Section 1.2).

Several mechanisms could, generally, explain health effects of early-life environmental exposures. At the systemic scale, interaction of stressors with the hormonal system (suggestive of endocrine disruption), the immune system (possibly leading to inflammatory responses) or the nervous system can play a role. An effect through an alteration of the microbiota also constitutes a (non-exclusive) plausible pathway [25]. At the molecular scale, epigenetic modifications could mediate part of the health effects of chemical stressors [5,26]. Assessment of the relation between epigenetic marks or gene expression in human studies in relation to exposures and health, an effort that has been termed *epigenetic epidemiology* [26], is developing in the context of early-life. Examples include characterization of the relations of DNA methylation with maternal active smoking [27,28], atmospheric pollutants [29,30] or endocrine disruptors [31]. Consideration of these and other biomarkers of effects such as hormonal levels, oxidative stress or immunological markers, which are common practice in toxicology, has now become possible in epidemiological studies through the collection of relevant biospecimens, corresponding to the advent of molecular epidemiology [32].

Thus, increasingly, epidemiological studies are capable to characterize not only the occurrence of adverse effects possibly induced by exposures, but also to point to the underlying mechanisms, which used to be a feature of toxicology alone. In spite of this increasing similarity in aims, toxicological and epidemiological studies are generally designed independently. This independent design tends to limit the overlap between these two approaches in terms of outcomes considered and, in general, limits comparability. A strong difference remains between both disciplines, that related to exposure characterization, since exposures are *observed* and generally not *controlled* in epidemiological studies. In order to efficiently identify if early-life exposures can alter the above-mentioned biological pathways and induce health effects, progress are required in the approaches used to assess exposures in epidemiology.

### 1.2. Issues Related to Exposure Assessment

Many of the above-mentioned factors for which moderate to strong evidence for health effects exists in humans (see Section 1.1 above) relate to exposures that can be quite efficiently assessed by the classical tools of (environmental) epidemiology: either questionnaires (e.g., tobacco smoke; the use of DES during pregnancy), biochemical assays based on spot biospecimens, for compounds with a long half-life in the human body (DDT, PCBs, to some extent perfluorinated compounds, although little accessible matrices such as fat tissue may be required), or outdoor environmental models (in the case of fine particulate matter). However, since persistent compounds are generally strongly regulated, most currently marketed chemicals are, as already mentioned, non-persistent. For example, the half-life of bisphenol A, DEHP or some organophosphate pesticides in the body is between a few hours and a few days; exposures, which may occur during meals, because of cosmetics use or through inhalation, are also likely to vary within and between days and weeks. Both features lead to very strong within-subject temporal variations in urine levels of the compounds or their metabolites [33,34,35]. For this reason, epidemiological studies of effects of short half-lived compounds are generally more challenging in terms of exposure assessment than studies of persistent compounds (although as already mentioned challenges also exist regarding the assessment of persistent compounds, e.g., in terms of identification of the most relevant matrix). Exposure measurement error is expected. Indeed, for these compounds with strong within-subject temporal variations, irrespectively of the accuracy of the biochemical assay, a spot biospecimen will only provide an estimate of exposure in the few hours before sample collection, while the toxicologically-relevant window may be much longer. In the context of classical type error, the probable error structure in biomarker-based studies, a strong attenuation bias in dose response functions is expected in studies relying on a spot biospecimen [33]. This bias towards the null may be as high as 80% (i.e., a slope of 2 observed on average if the true slope is 10) in the context of compounds with such strong within-subject variability as bisphenol A, whose intra-class coefficient of correlation (ICC) is about 0.2 [33]. Concomitantly, a strong decrease in statistical power is induced [33]. Such attenuation bias and the resulting power loss can hamper the development of a consistent epidemiological literature on the possible effects of non-persistent compounds, and might postpone regulatory actions important for public health.

Moving beyond the current state of the art requires progress in exposure assessment. One way forward is to assess exposures over temporal windows longer than a few hours or days; this can be achieved by relying on repeated assessment of exposures, or on collection of repeated biospecimens throughout the temporal window of interest within each subject. Following this logic, a *within-subject biospecimens pooling approach* allowing to efficiently estimate the urinary levels of biomarkers and metabolites with strong temporal variations has been developed. Several biospecimens are collected within each subject and pooled in each subject, following which the compound of interest is assessed in the pooled urine sample, providing an estimate of mean exposure in the whole time period of biospecimens collection. Compared to studies relying on a spot biospecimen, this approach allows to strongly limit bias and to increase power without increasing the assay costs [33]. The approach is supported theoretically [33] and empirically [36,37], and can be generalized to the context of multi-exposure (exposome) studies (Agier et al., unpublished data). Alternatively, exposure can be assessed in each of the repeated biospecimens, which, for an increased analytical cost, will also provide information on the within-subject temporal variability in exposure.

Only very seldom did studies rely on repeated exposure assays during or after pregnancy [38,39,40]. To limit bias due to exposure measurement error to less than 10%, collecting about half a dozen urine samples per subject during the relevant exposure window (e.g., pregnancy) may be required for compounds with an ICC of 0.6, while, for more variable compounds (ICC around 0.2), 35–40 samples per subject may be required [33]. To our knowledge, no large-scale epidemiological study on the health effects of endocrine-disruptors has collected such a large number of urine samples per subject.

Regarding the assessment of exposure to air pollution, outdoor models predicting exposures at the home address have been widely used [24]. However, pregnant women do not spend all their time at home; in addition, in industrialized countries, indoor levels of specific pollutants such as particulate matter are sometimes poorly correlated to personal exposure [41,42]. The longer autonomy of GPS (Global Positioning Systems) devices now allows to conveniently assess time-space activity [42,43]. Active and passive samplers with good accuracy can be carried by volunteers for several days [42,44,45], allowing to assess personal exposure to air pollution in different micro-environments. These tools have so far very seldom been used at a large scale in etiological studies in pregnant women [45,46,47,48]. Personal samplers are also particularly relevant for specific air pollutants whose outdoor levels are a very poor proxy of personal exposure. These include benzene, a recognized human carcinogen [49] for which few studies with efficient exposure assessment tools have been conducted outside the setting of occupational exposures, particularly in pregnant women [45].

### 1.3. Issues Related to Assessment of Health and Biological Parameters

Moving from exposures to their possible consequences in humans, challenges also exist when it comes to the assessment of effect biomarkers and of health. Notably, in relation to respiratory health, objective measures of lung function are usually only performed from the age of 5–6 years, when spirometry, which requires strong coordination and cooperation from the child, can be efficiently done. In the context of DOHaD hypothesis, it is relevant to aim for an earlier objective assessment of lung function. Relevant candidate approaches for this purpose include the *lung clearance index*, measured by multiple breath washout tests, which can be performed in infants during natural sleep, and the oscillometric measurement of the parameters of ventilatory mechanics, which can be assessed using the forced oscillation technique (FOT) in young children around three years. Both techniques allow assessing airway obstruction, which is associated with asthma. Such approaches have been so far little considered in cohorts of healthy children from the general population [50]. Similarly, regarding markers of effect, it is worth attempting assessing epigenetic marks in biologically-relevant tissues, such as the placenta [30,51], rather than in circulating blood, which provides distinct information [52]. Attempts to collect live cells in the context of human cohorts is also relevant and may allow characterization of their function and sensitivity to environmental factors.

### 1.4. Study Aims

We implemented these novel approaches in a new type of mother-child cohort [53] characterized by early recruitment during pregnancy, deep phenotyping and extended personal exposure characterization, called SEPAGES. SEPAGES (which stands for: Suivi de l’Exposition à la Pollution Atmosphérique durant la Grossesse et Effets sur la Santé; Assessment of air pollution exposure during pregnancy and effect on health in English) is a research platform in environmental health aiming to: (1) characterize finely exposures to ubiquitous components of the exposome during pregnancy and early life; currently funded projects in this cohort focus on two main large families of pollutants, atmospheric pollutants and non-persistent chemicals, including those with potential endocrine-disrupting properties, with the aim to broaden later on to the whole chemical exposome [54]; (2) evaluate the impact of the environment on mother-child health, focusing on three main outcomes: growth (foetal and child growth), respiratory health and neurodevelopment; (3) explore possible underlying biological pathways, focusing on those mediated by epigenetic marks, immunologic and hormonal parameters and by the gut microbiota.

Another originality of our approach is the combination of the human cohort with a toxicological study, whose design was developed together with that of the cohort. The toxicological experiment shared similarities in the exposures and outcomes considered, but differed from the cohort in that it also considered longer-term effects than the human cohort, such as effects on the offspring of the in-utero exposed generation, and also aimed to study mechanisms more finely [55]. Given the relatively large amount of studies on non-persistent chemicals in the toxicological literature, the animal study, a 2-generation monitoring of New-Zealand white rabbits, was focused on gestational exposure to air pollutants, specifically diesel engine exhaust [56]. This study and its first results were already presented elsewhere [56,57,58,59].

## 2. Design of SEPAGES Couple-Child Cohort

### 2.1. Study Population

#### 2.1.1. Study Area

The study area of SEPAGES cohort is centered around the Grenoble metropolitan area (population, 440,000) and has a size of 45 km by 90 km (about 4000 km^2^). Grenoble is a flat city located at 200 m above the sea level and surrounded by Alpine mountains, with a particular climate under continental, oceanic and Mediterranean influences and a strong thermal amplitude over the year. SEPAGES study population lives approximately in a buffer of 80 km around the centre of Grenoble, and includes urban, peri-urban and rural (including mountain) zones (Figure 1). The basin configuration, the relatively rare windy conditions, the reliance on old wood stoves for heating by a proportion of the population, concur to rather high concentrations of particulate matter, although not much higher than those of French cities of similar size. Mean (density-weighted) PM_2.5_ (PM with an aerodynamical diameter below 2.5 µ) concentration was 13.9 µg/m^3^ (5^th^–50^th^–95^th^ percentiles: 10.2–14.6–16.2 µg/m^3^) at the scale of the 49 cities of the urban area for the 2015–2017 period [60]. Contrasts of mean PM_2.5_ between the winter and summer seasons are high, as well as contrasts between Grenoble city and the surrounding area. The population has a rather high education level compared to the average French situation, due to the presence of a large higher education, research and engineering community.

#### 2.1.2. Recruitment

The study subjects of SEPAGES cohort were pregnant women, their partner and future child. Pregnant women had to fulfil the following eligibility criteria: being pregnant by less than 19 gestational weeks at inclusion, older than 18 years old, having a singleton pregnancy, planning to give birth in one of the four maternities clinics from Grenoble area and living in the study area. The fathers of the expected child were also offered to participate to the study, with no exclusion criteria. The participation of the father was not mandatory for the mother and the child to be included.

Recruitment took place between July 2014 and July 2017 in eight obstetrical ultrasonography practices located in Grenoble area. Information about the study was given directly by our fieldworkers (K.G., M.G.) to pregnant women generally coming for their 13-week ultrasound examination. This 13-week examination, when screening tests for trisomy 21 are performed, is part of the mandatory follow-up of pregnancies in France, and is generally planned some time ahead, which allowed SEPAGES fieldworker to target the ultrasonography practices where such visits were planned each day. Some women were approached while coming for an examination earlier during pregnancy, e.g., to confirm the pregnancy. Most (90%) included volunteers were recruited by a fieldworker, while the remaining 10% were informed and then recruited after reading a SEPAGES brochure or a poster in a medical center. All women with a 13-week ultrasonography appointment at a time when the fieldworker was present were approached by the fieldworker. The fieldworker administrated a short eligibility questionnaire to check the inclusion criteria and collect socio demographic information (age, level of education, occupation…) and describe women who refused to participate. Detailed information about the study protocol and a leaflet summarizing the protocol were given to the interested pregnant women who met the inclusion criteria. A few days later, the fieldworker called the woman to offer to participate and schedule the inclusion visit. After two calls, pregnant women who did not get back to the research team were sent up to two email reminders, after which we considered that the eligible woman had refused to participate.

Our fieldworkers approached 3360 women during the inclusion period, which, based on data from birth certificates covering Isère *département*, where Grenoble is located, corresponds to an estimated 14% of the pregnant women meeting SEPAGES inclusion criteria. The *eligibility rate* (proportion of eligible women among those approached) was 69%. Among eligible approached women, the *participation rate* was 21%. The final number of included families, corresponding to those who signed an inclusion consent form, was 484, while there were 471 families with at least one clinical examination and information about delivery (Figure 2).

Compared to pregnant women from France, pregnant women from Grenoble tend to be older, to have a lower parity and a higher education level (Table 1). The data collected in the recruitment questionnaire allowed to compare eligible women who participated with those who did not participate (Table 1). Compared with the approached women refusing to participate, participating women were older, had a lower parity, more often lived in a relationship and worked, and had a higher education level. Compared to pregnant women living in France, a significantly higher proportion of SEPAGES women had a body mass index (BMI) in the 18.5–24.9 kg/m^2^ range (71%, compared to 61% in France), did not smoke before pregnancy (89%, compared to 70% in France) or during pregnancy (93%, compared to 83% in France) and had to use an infertility treatment to become pregnant (10%, compared to 7% in France). A description of the children is given Table 2.

### 2.2. Pilot Study

A pilot study was conducted in 2012–2014 to validate the study protocol, in which 40 volunteers living in Grenoble area were included and followed-up until the child was one year old. The pilot study allowed to define the recruitment strategy, to assess the participation rate, the feasibility of the field work, the intensive urine collection protocol [61], the possibility to use personal air samplers [42], and to finalize the clinical examinations and biological collection protocols. The pilot study also allowed to compare and develop approaches for exposure assessment to air pollutants [42] and non-persistent chemicals [36], as well as to characterize the temporal variability of some of the chemical exposures of interest [61]. In what follows, we only present the final study. Details on the pilot study can be found elsewhere [36,42,61].

### 2.3. Biological Samples

During pregnancy, women underwent three (for the first 111 recruited women) or two (for the remaining women) follow-up weeks, during which biological specimens were collected. Visits took place in median at gestational week 18 (1^st^ visit; 25^th^–75^th^ centiles, 16–19), 26 (intermediate visit; 25^th^–75^th^ centiles, 25–27) and 34 (final visit; 25^th^–75^th^ centiles, 32–35). Biospecimens were also collected for the father and the child (Table 3).

#### 2.3.1. Urine Samples

Women collected samples from three micturitions per day during each follow-up week. From the population of SEPAGES feasibility study, we demonstrated that, when it comes to estimating the effects of exposures averaged over windows of a week or more, collecting three urine samples per day is not expected to entail bias, compared to a collection of all daily urine samples (which is more cumbersome); this applies even for compounds with strong within-subject (temporal) variability, such as bisphenol S [36]. In children, urine samples were collected daily during a week at 2, 12 and 36 months of age, using a cotton inserted in the diaper for the children who were not toilet-trained. Collection tubes (Sarstedt, Nümbrecht, Germany) guaranteed to be phthalate-free were provided to participating women. Women were asked to fill in a diary indicating the hours of all micturitions and delay before freezing. After collection, samples were stored at −20 °C at the house of the participants. At the end of the follow-up week, samples were picked up by a study fieldworker and brought to the biobank, stored at −20 °C until they were thawed at 4 °C, aliquoted and stored at −80 °C.

Following the within-subject biospecimens pooling approach [33,36], weekly pools aliquots (pools of the same volume of each of the samples collected over a follow-up week) were prepared for each subject, while daily pools and spot urine samples were kept for a subsample of volunteers only, to limit the number of stored biospecimens (Table 3).

#### 2.3.2. Blood Samples

Maternal blood was collected at home around 19 gestational weeks; paternal blood was collected during the inclusion visit in median at 30 gestational weeks, and child blood was collected at the hospital at birth (newborn blood spot on Guthrie cards and cord blood), and at one and three years by experienced personal.

#### 2.3.3. Biological Samples at Delivery

Participating women were given a collection kit to bring to the maternity clinic at the time of delivery. The kit contained instructions (which had previously been circulated to all participating clinics) and material for the collection of all samples to be collected at delivery, including placental tissues, cord blood, breast milk, child hair and meconium. Three placenta samples were immersed immediately by the clinic midwife in a RNA stabilization solution (RNAlater TissueProtect Tubes, Qiagen, Hilden, Germany) for 24 h before being transferred in a dry tube, which was then frozen at −80 °C. One placenta sample was immersed in a paraformaldehyde solution before being embedded in paraffin. Up to 40 mL of cord blood were drawn, of which 3 mL were put in an RNA stabilization solution tube (Tempus™ Blood RNA Tube, Life Technology, Carlsbad, CA, USA).

After collection, delivery biological samples were stored in the refrigerator at the maternity at +4 °C. A bicycle courier visited each maternity clinic every morning to pick up the samples and bring them in an ice bag to the biobank, where samples were aliquoted and frozen at −80 °C. An additional newborn blood spot was collected at day 3 in the framework of the national newborn thyroid hormones screening program.

#### 2.3.4. Fecal Samples

Fecal samples were collected up to four times during the first three years of life of the child (2 months, one year, two years (for a subsample) and three years). These samples are being used for the assessment of the gut microbiota composition through 16S ribosomal RNA gene sequencing, in Micalis Institute (Dr. P. Lepage, INRA, Jouy-en-Josas, France). Metagenomics assessment is also planned in a subgroup.

#### 2.3.5. Other Biospecimens

Buccal swabs were collected from each child at one and three years by scraping the inside of the cheek with a small plastic device. One swab was collected from each cheek; swabs were stored in cryovials. One vial was filled with 0.5 mL PBS (left cheek) while the other was filled with 0.5 mL RNA cell protect (right cheek).

Nasal swabs were collected from each child at three years. These samples were collected by rubbing one swab gently against the inner wall of each nostril, while getting a sample of mucus, if the child had a runny nose at that time. The swab was stored in a cryovial filled with transport medium (glycerol 15% in Iscove’s Media). These samples are meant to allow assessment of the airway microbiome [63].

Fingernails were also collected at home by a parent from each child at three years.

All samples except the hair, stored at room temperature, and the nails, stored at −20 °C, have been stored in Grenoble University Hospital certified biobank at −80 °C. Further details regarding the use of biospecimens for exposure assessment are given in the Section 2.4 and Section 2.5. below.

### 2.4. Assessment of Exposure to Environmental Pollutants

The environmental factors of interest include components of the outdoor exposome (particulate matter, nitrogen dioxide, benzene, toluene, ethylbenzene, xylenes, temperature), lifestyle factors such as diet and components of the “internal” exposome assessed from biospecimens (phenols, phthalates, DINCH metabolites, perfluorinated compounds, organophosphate pesticides in a subgroup…). The compounds for which assessment has already been done or is already funded are listed Table 4.

#### 2.4.1. Outdoor Exposome

##### Air Pollutants

The data collected will in particular allow to estimate (1) the average (outdoor) air pollution (PM_2.5_, PM_10_, nitrogen dioxide) exposure of the participant at the home address; (2) the outdoor exposure taking into account time-space activity [42], both during pregnancy and for the child; (3) PM_2.5_ indoor levels at the home address in a subgroup of 80 participants; (4) personal exposure to PM_2.5_, PM_2.5_ oxidative potential, nitrogen dioxide (for both mother and child), specific volatile organic compounds (benzene, toluene, ethylbenzene, xylenes, for mothers only) and soot (in a subgroup of mothers) from personal dosimeters (Figure 3a).

Residential addresses before, during and after pregnancy as well as addresses of maternal work place, kindergarten, school and any place where the child spends a significant amount of time were collected prospectively and geocoded. Outdoor levels of NO_2_, PM_10_ and PM_2.5_ in Grenoble urban area were assessed at fine spatial (10m)-temporal (hour) resolution using a dispersion modelling approach (relying on data on emission, meteorology and permanent monitoring stations) implemented by Atmo Auvergne-Rhône-Alpes, the regional air pollution monitoring network [60,64]. During each follow-up week, measurement devices (Table 4) were given to the participants at day 0 and then turned off and picked up at day 8. Personal devices measuring PM_2.5_ mass concentration (Micropem active air sampler, RTI, Research Triangle Park, NC, USA), soot (MicroAeth, AethLabs, San Francisco, CA, USA), NO_2_ (passive sampler from Passam AG, Männedorf, Switzerland), volatile organic compounds from the BTEX family (benzene, toluene, ethylbenzene and xylenes, using a passive air sampler from Passam AG) were placed together with a smartphone (Galaxy SIII, Samsung, Seoul, South Korea) on which an application was installed to collect geo-located data (ExpoApp, Ateknea Solutions, Barcelona, Spain) [65] in backpacks carried by the participants. A subgroup of 49 participants also carried an accelerometer (ActiGraph, Actigraph Corp, Pensacola, FL, USA) on the waist [66]. Indoor concentrations of PM_2.5_ at the participants’ homes were assessed in a subgroup. Participants were instructed to leave devices on during all the measurement week and to keep the devices close to them, including at nighttime. Any deviation from the protocol had to be recorded in a diary. Data registered by the different devices were uploaded to a secured server, except for the passive samplers, which were sent to Passam AG laboratory for analysis. Active air samplers were then calibrated and reusable devices were used again for another measurement week. The participants had the possibility to switch off the GPS application at any time, and relevant ethical agreements for the use of GPS were obtained. The filters from the active samplers carried by the women and then the child were transferred to IGE Grenoble (G. Uzu, IRD, Grenoble, France), where the oxidative potential of PM was assessed using the DTT and AA acellular assays [67,68].

##### Other Components of the Outdoor Exposome

Along with the devices used to measure air pollutant exposure during the repeated measurement weeks, noise was assessed using *NoiseTube* application (Software Languages Lab, Vrije University, Brussel, Belgium) installed on the smartphone carried by participants. Temperature was measured every 15 minutes during the measurement week using a thermometer (Table 4 and Figure 3a). Data on outdoor levels of meteorological parameters (temperature, humidity, pressure) were recorded through Meteofrance monitoring network. Activity and sleep assessments were performed using an accelerometer (ActiSleep, from ActiGraph Corp.) at 2 months, one year (for a subsample of children, worn at the ankle) and three years (worn at the waist).

#### 2.4.2. Behaviors

##### Drug Use, Cosmetics, Cleaning Products

The volunteers were asked to take pictures (including the bar code) of the cleaning and cosmetic products used as well as of any drug (e.g., analgesics) taken during each follow-up week, using the camera of the smartphone provided to them. For 113 volunteers during pregnancy and for at least one follow-up week of the child, a specific Smartphone application (Cobanet, EpiConcept, Paris, France) was used by the participant each time a cleaning product was used; the application allowed to associate a picture of the barcode of the product with a short questionnaire on the use of the product (frequency of use, purpose of use, protection upon use, etc.).

##### Nutrition and Diet

During pregnancy, food frequency questionnaires were completed online by the volunteers at each trimester of the pregnancy. Women were also asked to fill in a diary indicating the hour of each meal or snack during follow-up weeks. During the first three years of life of the child, parents were asked to record the child food intake during three days (two weekdays and one week-end day) at 2, 12, 24 and 36 months, and food frequency questionnaires were asked at 3, 6, 12, 18 and 36 months.

#### 2.4.3. Internal Exposome

The first components of the internal exposome that were assessed in SEPAGES cohort are from the families of phthalates, phenols (using the pooled biospecimens) and perfluoroalkyl acids (from spot maternal serum samples; see detailed list in Table 4 and Figure 3b). In a subgroup of 49 women also part of HELIX early life exposome project, organophosphate pesticide metabolites were in addition assessed using the pooled biospecimens [40,69]. Additional components of the chemical exposome will be assessed as part of ATHLETE EU (H2020) newly funded exposome project.

### 2.5. Biological Parameters and Health Outcomes Assessed

The main biological and health parameters whose assessment was completed or is planned relate to respiratory health, neurodevelopment, growth, DNA methylation, gut and airway microbiota, immunological function and thyroid hormones (Table 5).

#### 2.5.1. Respiratory Health

Child respiratory health was assessed by questionnaires on respiratory symptoms and by measures of respiratory function at 6 weeks and three years. New technologies allowing to measure the lung clearance index (reflect of the inhomogeneity of the ventilation) and functional residual capacity were used at 6 weeks. The lung examination took place at the hospital during natural sleep of the child, without artificial sedation. A mask was applied on the newborn’s face and measurements were performed in quiet sleep using the Exhalyzer D (Ecomedics, Basel, Switzerland), a compact system incorporating all elements of a fully-equipped infant pulmonary function testing device, allowing non-invasive assessment of respiratory function. Two measures were performed:-Functional residual capacity (FRC), a measure of lung volume and lung clearance index (LCI), which reflects inhomogeneity of the ventilation. These were assessed by multiple breath washout (MBW) technique using pure oxygen as tracing gas. These parameters reflect lung physiology and are considered early predictors of adverse respiratory health in childhood [70]. The mean of two to three valid measures conducted within an interval of 10-15 minutes was recorded [50].-Tidal breathing flow-volume loops (TBFVL) in quiet sleep. One hundred cycles were recorded per child. The main parameters assessed were respiratory frequency, mean respiratory flow and ratio of peak tidal expiratory flow to total expiratory time (tPTEF/tE), a proxy of bronchial obstruction.

At three years, child respiratory function was assessed using the forced oscillation technique (FOT) [71] with the TremoFlo (Thorasys Systems, Montreal, QC, Canada), a non-invasive method allowing to measure airway resistance and reactance.

In addition, information on respiratory symptoms and diseases (including wheezing, bronchitis, bronchiolitis) during the three first years of life in the child were recorded by questionnaires to the parents. Both parents underwent spirometry (after delivery for the mother), an objective lung function test providing airflow and lung volume measurements.

#### 2.5.2. Growth

Foetal growth was assessed by ultrasonography measurements performed at 10–14, 20–24 and 30–34 weeks of amenorrhea in the framework of the normal pregnancy follow-up. Measurements included biparietal diameter, head circumference, abdominal circumference, transverse abdominal diameter and femoral length. Weight, height and head circumference of the child along with placental weight were measured at birth.

Weight, height and head circumference of infants, skinfold thickness and body impedance (only at 36 months, using BodyStat 1500 MDD, Bodystat, Isle of Man, UK) were measured in a standardized way during the 6-week, 12- and 36-month study examinations. Additionally, all weight, height and head circumference measurements performed by health care practitioners were copied from the child health booklet.

#### 2.5.3. Neurodevelopment

Child neurodevelopment was assessed longitudinally using both clinical assessments performed by trained SEPAGES fieldworkers, including trained neuropsychologists, and validated questionnaires completed by the parents. We assessed several dimensions of neurodevelopment, such as cognition, behavior and motricity. Questionnaires included the Vineland Adaptive Behavior Scales (VABS) [72] and the MacArthur-Bates Communicative Development Inventory at 1 and 2 years; the Child Behavior Checklist [73] at 2 years; the Behavior Rating Inventory of Executive Function-Preschool (BRIEF-P) and the Social Responsiveness Scale (SRS) at 3 years. At 1 year, social withdrawal was evaluated using the Alarm Distress Baby scale. At 3 years, cognitive function was evaluated using the fourth version of the Wechsler Preschool and Primary Scale of Intelligence.

In addition, a sub-sample of the cohort was included in an eye-tracking protocol. The protocol involved four tasks: scenes exploration, faces recognition, saccadic reflexes and smooth pursuit. The tasks targeted specific aspects of development, including orientation to social stimuli in natural scenes, development of visual attention to face, face recognition and development of oculomotor control via assessment of reaction time, saccadic response and smooth-pursuit.

The natural scene perception task is widely used in adults to model early attention to the “cognitive objects” present in scenes, including faces, animals and other signs of life [74]. This task is often used to test predictions from computational model of visual attention [75]. The face perception task was aimed both at quantifying the structure of infant’s exploration of static faces (i.e., attention to eyes) and the infant’s recognition of a particular face (through increased attention toward novel faces) (Figure 4). Recent eye-tracking results highlighted the value of orientation to eyes as an early predictor for atypical development, such as autism [76] and William’s syndrome [77]. The saccade-to-target task provided access to markers of oculomotor development such as saccadic reaction time and saccade kinematics. The smooth-pursuit task targeted aspects of predictive oculomotor control, including catch-up responses and pursuit-gain. The Eyelink1000 system used to track infants’ gaze (head-free setup) also provided data about head-turns and back and forth movements. These data were used to build a general description of infant motor activity. The eye-tracker data were moreover used to evaluate aspect of infants’ behavior across all tasks such as blinking rate or mean fixation duration. The oculometric data will help describing idiosyncratic aspects of infant development. The examinations took place at BabyLab center (Laboratoire de Psychologie et Neurocognition, University Grenoble-Alpes, CNRS).

#### 2.5.4. Methylome and Microbiome

DNA methylation was assessed at Centre National de la Recherche en Génomique Humaine (CEA, Dr. J. Tost) from the placental biopsies and maternal peripheral blood from the first study visit using Infinium CytoSNP-850K Beadchip (Illumina, San Diego, CA, USA). From the children fecal samples, as mentioned above, the gut microbiota will be assessed using 16S ribosomal RNA gene sequencing (Dr. Lepage, INRA). Nasal swabs are meant to allow to explore the airway microbiome.

#### 2.5.5. Thyroid-Related Hormones

Thyroid-stimulating hormone (TSH) and thyroxine (T4) were assessed in the newborn blood spot collected on a Guthrie card two to three days after birth. TSH assessments were done as part of the national screening program on congenital hypothyroidism, while T4 assessment was done on the remaining blood, if any. Assessments of TSH, T4 (free and total), T3 (free and total) and selenium in maternal serum along with iodine in maternal urine were performed for all SEPAGES women.

#### 2.5.6. Immunological Parameters

We analyzed phenotypic and functional features of the immune system, which included tests on heparinized whole blood fresh samples: (1) quantification of cell subsets and their activation state by flow cytometry and (2) analysis of the functionality of T cells and dendritic cells. Six hundred microliters of blood were necessary for these tests; the remaining sample was centrifuged to obtain plasma, and peripheral blood mononuclear cells (PBMC) were isolated from the diluted blood pellet by density gradient centrifugation. Plasma (2 mL) and PBMC (about 5 million cells) were stored frozen for further analyses. Samples from more than 300 maternal (at gestational week 19 in median) and 150 cord blood samples were processed.

Immunophenotyping was performed on fresh blood to quantify the percentages and absolute concentrations of most immune cell subsets: T cells (CD4pos, CD8pos, CD56pos, regulatory T cells, Th2 T cells), monocytes (CD16pos and CD16neg), B cells, natural killer cells (CD56high CD16low and CD56low CD16high), dendritic cells (plasmacytoid DC, myeloid DC (BDCA3pos and neg), granulocytes (neutrophils, eosinophils, basophils). This phenotype was assessed using a panel of antibodies in three tubes, with 8-color flow cytometer (table S1); CountBright™ Absolute Counting Beads (Molecular Probes, Eugene, OR, USA) were added to the tubes to define each subset concentration. The activation state of each cell subset was accessed by analysing the median fluorescence intensity of relevant activation markers (CD25 marker for T cells; CD40, CD86 for dendritic cells and monocytes; CD16 and CD66b on myeloid cells, and HLA-DR for all cell subsets).

Additionally, functionality of T cells and dendritic cells was evaluated by direct activation on whole blood with phytohemagglutinin (PHA, a mitogenic lectin targeting T lymphocytes) and R848 (resiquimod, a TLR7/8 ligand, triggering PDC and MDC activation) respectively. After 24 h of incubation, the supernatants were harvested and kept frozen for cytokine measurement. In PHA-activated samples, global T cell activity, Th1, Th2, Th17, Th9 and Treg polarization was evaluated by quantification of IL-2, TNFα, IFNγ IL-13, IL-17, IL-9, and IL-10. In R848-activated samples, global dendritic cells, PDC and MDC activity was evaluated by quantification of TNFα, IL-6, IL-8, IFNα, IL-1β and IL-12.

#### 2.5.7. Cardiovascular Health

The maternal cardiovascular function was assessed through blood pressure measurements at first and third trimester pregnancy visits and an electrocardiogram (ECG) at the first visit. Child blood pressure was assessed at the calf at 6 weeks (in a subgroup of children), one and three years (at the arm) using a T105S device (Omron Healthcare, Kyoto, Japan). Finally, our fieldworkers also assessed ano-genital distance at the age of 6 weeks using a standardized protocol [78].

### 2.6. Other Covariates

From questionnaires at different time points we collected information on sociodemographic factors (including parental occupation), medical history, medical treatments used, health events, tobacco consumption, passive smoking exposure during pregnancy and postnatally, characteristics of the home, type of child day care, sleeping duration of mother and child.

### 2.7. Overview of Data Collection, Storage and Management System

A specific informatic platform was designed for the cohort, allowing (1) questionnaire implementation by the SEPAGES team, (2) filling of online questionnaires by study participants and fieldworkers, (3) automatic text messages and emails notifications to the volunteers (Figure 5).

Most data related to SEPAGES cohort, including all individual information, are stored on secured servers from Inserm RE-CO-NAI platform (Villejuif, France), which meet the security criteria to host health data. A query-enabled data infrastructure allows to extract easily any type of data stored on the platform.

### 2.8. Secured Data Linkage

To ensure a high level of data security, a data linkage procedure was developed and set up by Epiconcept (Paris). A 13-character identifier was generated for each questionnaire, biological sample, health and exposure data, so that two pieces of information from a given participant are not identified by the same code. Consequently, each subject had between 100 to 200 different identifiers codes. A secured linkage table allows to link all data related to a participant.

### 2.9. Ethical Agreements

All mothers and fathers of the expected child signed an informed consent form for themselves and their child. Participants were given the possibility to accept or refuse some parts of the study, such as genetic analyses, biological samples collection and geo-localisation. All consent forms were signed by both the participant and one of SEPAGES medical investigators.

The ‘promoter’ of SEPAGES study is Grenoble-Alpes University Hospital (PI, R. Slama, I. Pin). Ethical agreements were obtained from the CPP (Comité de Protection des Personnes Sud-Est V) and the Commission Nationale de l’Informatique et des Libertés (CNIL), the French data privacy institution.

## 3. Conclusions

### 3.1. A New Type of Cohort with Intense Exposure Assessment

From previous studies, it was shown that some approaches used in the past in epidemiology to characterize exposures to specific factors strongly suffer from exposure measurement error, which can in turn lead to bias in dose-response functions and impact statistical power [33,36,37,79,80]. In particular, personal exposure to fine particulate matter is generally little to moderately correlated to outdoor levels [41,42], and the situation is similar for nitrogen dioxide, while in the case of airborne pollutants such as benzene and other volatile organic compounds, outdoor models probably represent an even poorer exposure proxy. Note that in the case of airborne pollutants, a part of this error may have a Berkson component, which is not expected to strongly impact dose-response function estimates, but can influence their estimated uncertainty and thus statistical power. Regarding non-persistent compounds assessed from exposure biomarkers, we and others have documented that relying on a spot biospecimen to assess exposures, as done in most previous human studies, is likely to entail exposure misclassification because of the strong within-subject variability of urinary concentrations [61,81,82]. As a result, attenuation bias in dose-response function, typically by as much as 20 to 80% for compounds with an intra-class coefficient of correlation in the 0.2–0.8 range as is the case for many currently marketed chemicals (e.g., members of the phenols, phthalates or organophosphate pesticides families) is expected [33]. Collecting about 40 urine samples per subject as done in our cohort is expected to limit attenuation bias in dose-response functions to less than 10%, even for compounds with an intra-class coefficient of correlation as low as 0.2 [33], making the usual assumption that biospecimens are collected during the toxicologically-relevant exposure window.

It can be argued that personal exposure estimates are sometimes more prone to residual confounding than proxy (e.g., questionnaire-based) exposure estimates [83]. This may for example arise because specific genetic polymorphisms may influence both the biomarker level and the risk of the disease considered. However, to our knowledge, there are very few, if any, efficient alternatives to biomarkers for most of the chemicals we are interested in, which have multiple sources (diet, cosmetics, ambient air…) about which questionnaires are of very limited help because subjects simply do not know which chemicals are in their environment. Regarding fine particulate matter and NO_2_ exposures, personal dosimeters such as those used in SEPAGES cohort allow to provide a better estimate of personal exposure during the follow-up weeks, although this is achieved at the cost of a decrease in the temporal resolution of the exposure estimates because dosimeters could not be carried during the whole pregnancy and childhood but only during the follow-up weeks. Of course, they take into account other sources than environmental (outdoor) models; for example, personal PM_2.5_ exposure estimates include exposure from cooking and tobacco smoke (although in the latter case, active and passive smoking have a very low prevalence in our cohort).

### 3.2. Overview of First Findings and Analyses Currently Planned

The first results of SEPAGES cohort study, based on a subgroup of our population pooled with two other equal-sized groups from Barcelona and Oslo as part of the Helix project, considered blood pressure during pregnancy in relation with phenols such as bisphenol A, phthalates and organophosphate pesticides pregnancy levels assessed from pooled samples [40].

From a practical perspective, the SEPAGES cohort provides a demonstration of the practical feasibility of deep phenotyping and exposure characterization in pregnant women and children from the general population. Indeed, compliance to the use of personal dosimeters for repeated 1-week periods and repeated urine collection was very good. The repeated biospecimens will allow reliance on the within-subject biospecimens pooling approach, a method whose theoretical validity was previously demonstrated [33,36] but which, to our knowledge, had so far never been used at such a large scale. If women collect an identical number of biospecimens each, statistical analyses relating exposures assessed by within-subject pooling and biological parameters can be done by regression modeling like in the case of exposures assessed from spot biospecimens [33], or even in a simpler way, as correction for biospecimens’ sampling condition [84] may not be required. That is, the exposure estimate from each (e.g., weekly) within-subject urine pool can be separately related to the outcome of interest in a (e.g., linear) regression model, adjusting for the relevant factors. If between-subject variations in the number of biospecimens collected exist (these were very low between SEPAGES women, most of which collected the planned number of biospecimens), Perrier et al. suggested to apply weights to the regression models, whose value depend on the number of biospecimens collected in each subject [33].

The overall aim of the cohort is to provide an improved characterization of the effect of time-varying prevalent environmental factors on continuous health and biological parameters related to respiratory health, growth and neurodevelopment, as well as to contribute to the unraveling of the underlying biological pathways possibly implied. The first analyses planned include: (1) the description of lung function trajectories in early-life; (2) the study of influences of air pollutants (PM_2.5_, NO_2_, benzene) on foetal growth, lung function and placental epigenetic marks; (3) the study of possible effects of exposure to endocrine disruptors on the same outcomes as well as on child’s neurodevelopment and gut microbiota. Projects related to the effects of temperature on health and epigenetic marks, as well as of the epigenome as a whole are also envisioned.

### 3.3. Strengths and Weaknesses of the Cohort

The main strengths of SEPAGES cohort are the deep phenotyping and characterization of exposures and the very rich biobank, currently including over 65,000 biospecimens from about 480 couple-child triads. Exposure assessment follows a so far unique and intense protocol starting in the first half of pregnancy and repeated throughout pregnancy and the first three years of the child’s life. The extensive and very rich biological collection (including urine, blood, placenta, feces, meconium, live cells, hair, nails) and the early lung function assessment at 6 weeks and 3 years of age are other strong assets, which will allow studying the possible effects of environmental factors on relevant biological pathways. Health outcomes are also assessed repeatedly, at the ages of 6 weeks, 1 and 3 years. Another strength of the cohort is the collection of data on the father of the child, through a clinical examination, biological samples and questionnaires. Finally, the monocentric nature of the study is key when it comes to ensuring a high homogeneity of data collection.

A weakness is that the sample size of the study is too small to study rare health effects without pooling with other cohorts (which is planned as part of ATHLETE EU exposome project). Most of the outcomes for which power is expected to be high enough considering SEPAGES alone are continuous outcomes, such as growth, lung function parameters and neurodevelopment assessed on quantitative scales rather than binary clinical outcomes. Moreover, due to the intensive protocol, well-educated participants are over-represented in the cohort, compared to the regional or national populations. This feature, which is shared by most environmental health cohorts, will impede studying if effects of specific exposures differ with sociodemographic categories, which is not an aim of the study. Lack of representativeness is, generally, not a limitation in etiologic studies, as discussed elsewhere [85,86]. Indeed, it is not because the exposure distribution is not representative of that of the source population that one can expect a bias in dose-response functions. Moreover, this over-representation should not strongly impact the exposure contrasts in the population; indeed, not all exposures have very strong sociodemographic gradients. Interestingly, focusing on a more homogeneous population like done in SEPAGES can limit confounding bias by some factors with strong health effects. For example, with a prevalence of active smoking during pregnancy of only 7% (smoking any time during pregnancy) and of obesity of 4% in our cohort, compared to 17% (smoking during third trimester of pregnancy) and 12% (obesity) at the national level [62], any confounding bias from these factors will be attenuated, compared to what would be observed in a representative population, assuming that recruiting such a representative sample while accurately assessing exposures is feasible.

The protocol presented here constitutes what one can call a third generation of mother-child cohorts (Table 6 and [53]), with early recruitment in pregnancy, personal air pollution dosimetry, repeated collection of urine samples for an accurate characterization of exposure to non-persistent chemicals, collection of a large number of biospecimens including placenta (DNA, RNA, tissue), meconium and buccal cells for microbiota assessment, and effort to conduct a toxicological experiment on a relevant animal model in parallel to the cohort. We believe that this approach can strengthen the level of evidence regarding the health and biological (e.g., on epigenetic marks or on the microbiota) effects of early life environmental exposures such as non-persistent endocrine disruptors of the phenols and phthalates family, air pollutants with strong temporal variations including fine particulate matter, nitrogen dioxide and benzene and other components of the exposome that could be assessed from the stored samples.

### 3.4. Mode of Collaboration and Existing Collaborations

A collaboration exists between HELIX early-life Exposome project [40,90] and SEPAGES cohort, with 49 SEPAGES volunteers being part of HELIX pregnant panel group; collaborations also exist with other research teams implied e.g., in microbiota, air pollution or sleep research, and a collaboration will be set up as part of the newly funded ATHLETE exposome project (funding, EU, DG Research, H2020 program). SEPAGES steering committee evaluates proposals from other research teams for relevance.

### 3.5. Epidemiology and Toxicology Joining Forces for DOHaD Research

An originality of our approach lies in the development of a toxicological experiment simultaneously to the development of the development of the cohort protocol. These two studies were meant to overlap in terms of characterization of the effects of early-life (intra-uterine) exposure to atmospheric pollutants, for which the toxicological literature is relatively scarce [11,12], possibly because of the technical challenges of inhalation as an exposure route, compared to food or intravenous exposure. Regarding non-persistent chemicals such as those from the phenols or phthalates families, the effects of intra-uterine exposure in animal models has been comparatively much more studied [91]. The first findings from the toxicological experiment dealt with the transplacental transfer of nanoparticles, their presence in olfactory neurons in the F1 generation [59], the link between diesel engine exhaust exposure and placental vascular function, growth and cardiometabolic parameters of the F1 and F2 generations [56], olfactory function [59], fatty acid profile in blood and placenta of the F2 generation [57] as well as semen parameters in the F1 generation, where a possible increase in sperm DNA fragmentation rate was suggested [58].

The toxicological approach typically allows going more in depth than in an epidemiological setting, by tackling a large number of physiological systems and investigating more deeply the biological mechanisms underlying the effects of exposures. The complementarity of both approaches is embodied in the Russo-Williamson thesis, according to which the establishment of causal claims in medicine require both *probabilistic* (difference in disease rates between exposure groups) and *mechanistic* evidence [92]. These distinctions between disciplines tend to attenuate, with epidemiology being increasingly capable to investigate mechanisms, or at least to point to possibly implied biological pathways through biospecimens collection and decreasing costs of molecular assays; a limitation is that, besides the placenta, biospecimens can be collected in few organs in healthy populations.

So far, to our knowledge, very few epidemiological and toxicological studies have been designed and conducted together. Apart from the studies presented here, interesting exceptions exist. For example, Heinrich et al. conducted a series of studies describing the association of air pollution levels with respiratory health in populations from Eastern Germany [93], and assessed the effects of particles sampled in the atmosphere of these areas on mice using an experimental design [91]. Regarding pesticides, Fitzmaurice et al. identified mechanisms of action of the fungicide benomyl on Parkinson disease using in vitro and in vivo assays, while an epidemiological study documented the association of occupational benomyl exposure on Parkinson disease incidence [94]. In relation to drugs exposure, an original approach allowed to tackle the issue of the effects of pregnancy use of analgesics on male reproductive disorders using both an observational human pregnancy cohort and an experimental approach in rats [95].

Of course, epidemiological and toxicological studies take place continuously and can be incorporated into the synthesis of the evidence regarding the health effects of pollutants even if they were not designed simultaneously. However, it is our experience that the collaboration between scientists from both disciplines has added benefits; it allows a better understanding of the strengths and challenges of each approach and facilitates the dialogue between the disciplines; it also enables the studies from the respective disciplines to have increased comparability. In the case of our study, the input from epidemiologists on the human evidence in favor of effects of atmospheric pollutants on foetal growth, birth weight and head circumference [24] led to effort to monitor foetal growth and head circumference in the toxicological study presented here [56] (something to our knowledge little done in the previous toxicological studies on the topic). The effects of atmospheric pollutants on placental histology identified by the present toxicological study [56] and previous ones [11] motivated the human study to assess placental histology. This parallel between both approaches also has limitations. For example, the possible effects of diesel engine exhaust on triglyceride levels and metabolism highlighted by the experiment in the F2 generation [56] cannot be confirmed on the short term in the human cohort; this might be possible in other existing human populations with long follow-up of families and assessment of air pollution in the grandparents. This illustrates the complementarity of the toxicological and epidemiological approaches and the promises of closer collaborations for improved evaluation of the health effects of environmental pollutants. This interdisciplinary logic is in line with the development of *weight of evidence*-type approaches in risk assessment [96]. 

## Figures and Tables

**Figure 1 ijerph-16-03888-f001:**
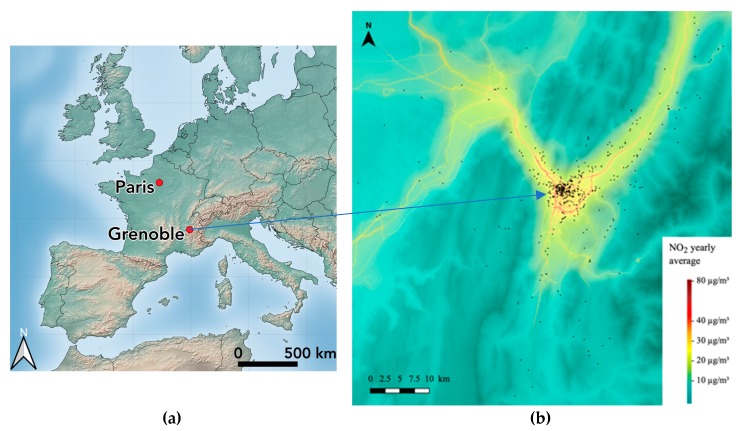
(**a**) Map of Europe and (**b**) of the cohort study area, indicating the yearly NO_2_ level (µg/m^3^, 2016) superimposed with the volunteers’ home addresses. Source: Atmo Auvergne Rhône-Alpes.

**Figure 2 ijerph-16-03888-f002:**
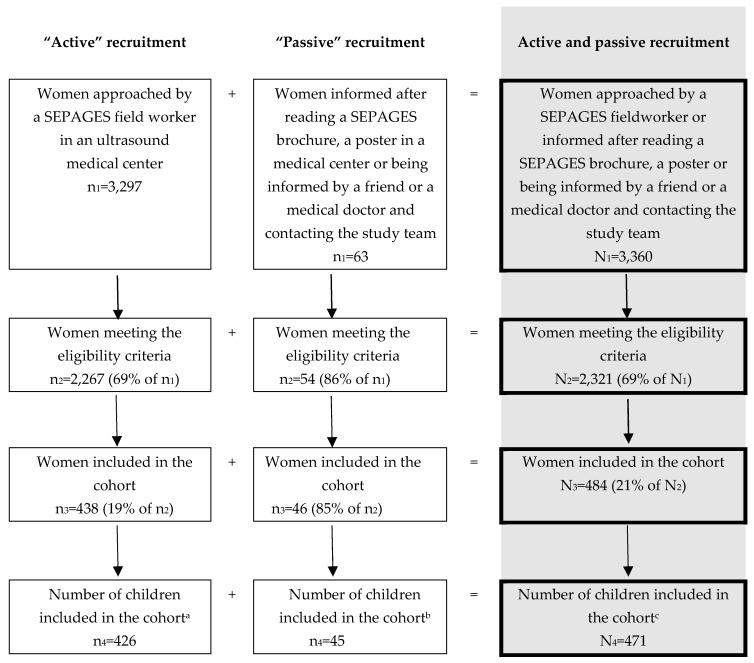
Flow chart of the recruitment of pregnant women in SEPAGES cohort. a: 12 women dropped out during pregnancy; b: 1 woman dropped out during pregnancy; c: 13 women dropped out during pregnancy.

**Figure 3 ijerph-16-03888-f003:**
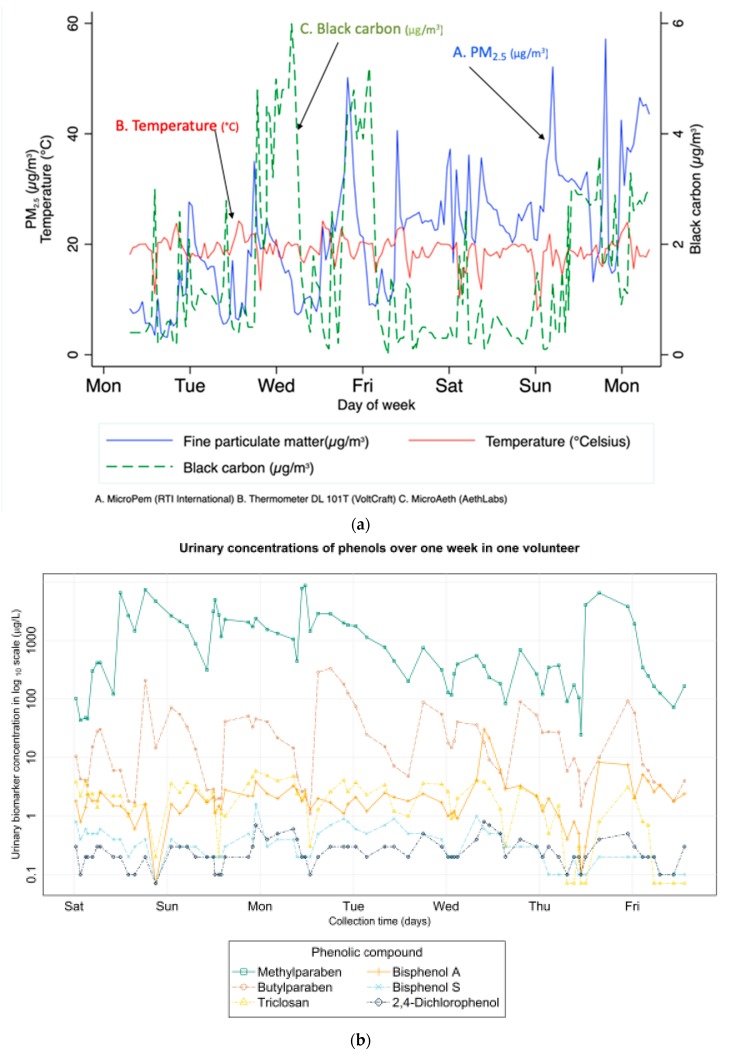
(**a**) Temporal variations of personal exposures in one follow-up week in one pregnant woman from SEPAGES cohort. (A) PM_2.5_ concentration, µg/m^3^; (B) temperature, °C; (C) Black carbon concentration, µg/m^3^. (**b**) Variation of urinary concentrations of phenols in urine samples collected during one week (one pregnant woman from SEPAGES-feasibility cohort; see Vernet et al. [61]).

**Figure 4 ijerph-16-03888-f004:**
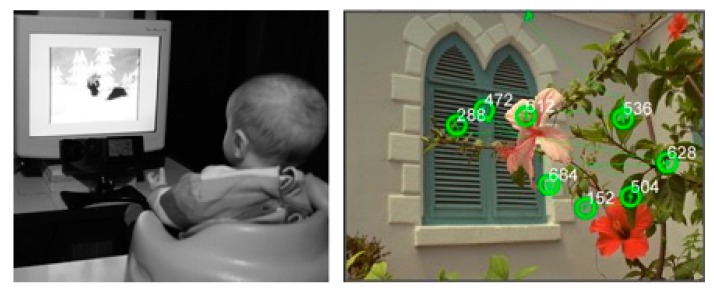
Evaluation of the eye-tracking protocol with a 6-month-old child; illustration of the setup and data in the spatial domain. Left: preparing for calibration. Right: scan path of the eye trajectory (numbers indicate fixation duration, in milliseconds).

**Figure 5 ijerph-16-03888-f005:**
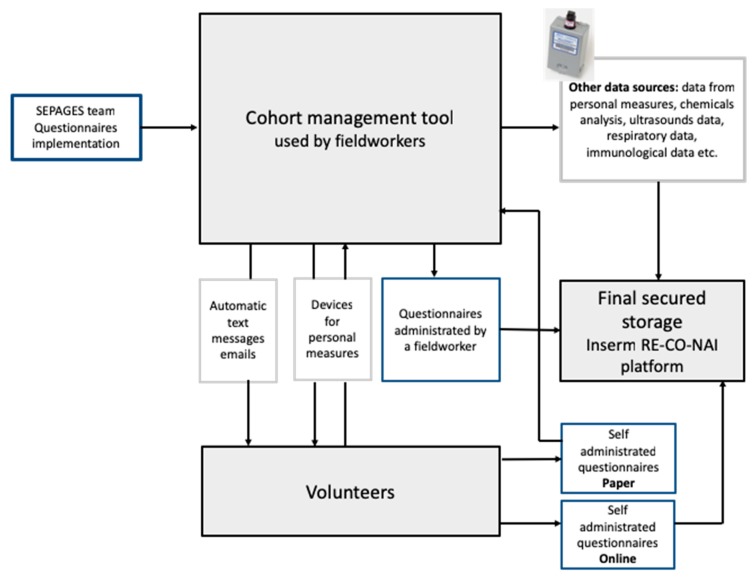
Data collection and storage platform structure for a paper-free cohort.

**Table 1 ijerph-16-03888-t001:** Description of SEPAGES cohort women and comparison with samples of pregnant women from Grenoble and France.

Characteristic	Population of Pregnant Women							
	SEPAGES Women *n* = 484		Approached but Not Included ^1^ *n* = 1841		Whole Grenoble Area ^2^ *n* = 17,899		Whole France ^3^ *n* = 12,950	
Age (years), mean ± SD	32.7 ± 3.9		31.0 ± 4.7		31.1 ± 5.0		30.3 ± 5.2	
Age (categories)				(<0.001) ^4^		(<0.001) ^5^	(<0.001) ^6^	
<20	0	(0.0)	13	(0.7)	113	(0.6)	204	(2.5)
20–24	13	(2.7)	128	(7.0)	1483	(8.3)	1553	(12.0)
25–29	113	(23.3)	589	(32.0)	5116	(28.6)	4052	(31.3)
30–34	230	(47.5)	683	(37.1)	6656	(37.2)	4377	(33.8)
35–39	117	(24.2)	353	(19.2)	3626	(20.3)	2236	(17.3)
≥40	11	(2.3)	74	(4.0)	900	(5.0)	519	(4.0)
Maternal Parity ^7^			(0.002) ^4^		(<0.001) ^5^		(<0.001) ^6^	
0	222	(45.9)	816	(44.6)	6036	(39.7)	5464	(42.2)
1 child	214	(44.2)	721	(39.4)	6098	(40.1)	4609	(35.6)
≥2 children	48	(9.9)	294	(16.1)	3071	(20.2)	2872	(22.2)
Marital status			(0.005) ^4^				(<0.001) ^6^	
In a relationship ^8^	483	(99.8)	1808	(98.2)	NA		9593	(81.9)
No relationship	1	(0.2)	33	(1.8)			2123	(18.1)
Education level			(<0.001) ^4^		(<0.001) ^5^		(<0.001) ^6^	
Primary school	0	(0.0)	4	(0.2)	106	(1.4)	187	(1.6)
Secondary Education	6	(1.2)	226	(12.3)	677	(9.2)	2489	(21.3)
High School education (Bac)	23	(4.8)	316	(17.3)	1404	(19.2)	2521	(21.6)
Undergraduate or graduate	452	(94.0)	1285	(70.2)	5141	(70.2)	6464	(55.4)
Nationality							(<0.001) ^6^	
French	394	(94.7)	NA		NA		10,083	(85.9)
Other European country ^9^	18	(4.3)					416	(3.5)
African country	0	(0.0)					993	(8.5)
Other nationality	4	(1.0)					243	(2.1)
Working status during pregnancy			(<0.001) ^4^		(<0.001) ^5^		(<0.001) ^6^	
Employed	434	(92.9)	1532	(85.0)	6806	(75.0)	7830	(68.1)
Unemployed	13	(2.8)	79	(4.4)	508	(5.6)	1928	(16.8)
Housewife/parental leave/in training	20	(4.3)	191	(10.6)	1757	(19.4)	1630	(14.2)
Not working, other	0	(0.0)	0	(0.0)	NA		108	(0.9)
Infertility treatment							(0.02) ^6^	
None	426	(90.1)	NA		NA		10,896	(93.1)
ART ^10^, ovulation induction	47	(9.9)					805	(6.9)
Height							(0.80) ^6^	
<160 cm	87	(18.1)	NA		NA		2,206	(18.9)
160–169 cm	281	(58.5)					6744	(57.8)
170–179 cm	105	(21.9)					2587	(22.2)
≥180 cm	7	(1.5)					121	(1.0)
Weight before pregnancy							(<0.001) ^6^	
<50 kg	44	(9.1)	NA		NA		968	(8.3)
50–59 kg	205	(42.4)					3791	(32.5)
60–69 kg	147	(30.4)					3424	(29.4)
70–79 kg	59	(12.2)					1816	(15.6)
≥80 kg	29	(6.0)					1661	(14.2)
BMI before pregnancy							(<0.001) ^6^	
<18.5 kg/m^2^	29	(6.0)	NA		NA		863	(7.4)
18.5–24.9 kg/m^2^	364	(75.8)					7045	(60.8)
25–29.9 kg/m^2^	67	(14.0)					2312	(20.0)
≥ 30 kg/m^2^	20	(4.2)					1368	(11.8)
Smoking before pregnancy							(<0.001) ^6^	
0	385	(89.1)	NA		NA		8217	(69.5)
1–9 cig./day	37	(8.6)					1350	(10.9)
≥ 10 cig./day	10	(2.3)					2132	(19.6)
Smoking during pregnancy ^11^							(<0.001) ^6^	
0	402	(93.3)	NA		NA		9798	(83.4)
1–10 cig./day	29	(6.7)					1447	(12.3)
>10 cig./day	0	(0.0)					499	(4.2)

Values reported are numbers (%), unless stated otherwise. BMI: Body Mass Index. ^1^ Pregnant woman interviewed by a SEPAGES fieldworker in an ultrasound medical center who met the SEPAGES inclusion criteria and did not want to participate to the study. ^2^ Database of birth certificates provided 8 days after birth and covering Isère *département*, where Grenoble is located. The population was restricted to women (1) who gave birth in one of the 4 maternity wards of Grenoble area, (2) who were older than 18 years old when they gave birth and (3) whose date of last menstrual period was between March 2014 (to match with the SEPAGES population) and February 2017 (no data were available after this date). ^3^ Source: 2016 French Perinatal Survey [62]. ^4^ P-value; chi-square test (or Fisher exact test when needed) comparing the characteristics of pregnant women included in SEPAGES and the pregnant women not included in SEPAGES and interviewed by a SEPAGES fieldworker in an ultrasound medical center. ^5^ P-value; chi-square test (or Fisher exact test when needed) comparing the characteristics of pregnant women included in SEPAGES and pregnant women living in Grenoble area. ^6^ P-value; chi-square test (or Fisher exact test when needed) comparing the characteristics of pregnant women included in SEPAGES and pregnant women living in France (2016 French Perinatal Survey [62]). ^7^ Before the index pregnancy. ^8^ Cohabitation or married. ^9^ Including Turkish. ^10^ Assisted Reproduction Technology. ^11^ For the pregnant women included in SEPAGES, smoking during pregnancy was defined as smoking any time during pregnancy. For the pregnant women living in France, smoking during pregnancy was defined as smoking during third trimester of pregnancy.

**Table 2 ijerph-16-03888-t002:** Description of the children from SEPAGES cohort, and comparison with newborns from Grenoble and France.

Characteristic	Children Population					
	Included in SEPAGES (*n* = 471)		Grenoble ^1^ (*n* = 17,899)		Whole France ^2^ (*n* = 13,158)	
Sex			(0.18) ^3^		(0.02) ^4^	
Girl	218	(46.5)	8878	(49.7)	6630	(52.0)
Boy	251	(53.5)	8986	(50.3)	6118	(48.0)
Gestational duration				(0.05) ^3^		(<0.001) ^4^
≤37 weeks of amenorrhea	50	(10.6)	1908	(10.7)	1938	(14.7)
38–39 weeks of amenorrhea	181	(38.4)	7919	(44.5)	5593	(42.5)
40 weeks of amenorrhea	146	(31.0)	4793	(26.9)	3348	(25.4)
≥ 41 weeks of amenorrhea	94	(20.0)	3194	(17.9)	2277	(17.3)
Weight at birth			(0.41) ^3^		(<0.001) ^4^	
<1500 g	2	(0.4)	157	(0.9)	140	(1.1)
1500–2499 g 2500–2999 g	14 80	(3.0) (17.2)	740 3346	(4.1) (18.7)	840 2716	(6.4) (20.6)
≥3000 g	369	(79.4)	13,629	(76.3)	9462	(71.9)
Length at birth			(0.09) ^3^		(<0.001) ^4^	
≤47 cm	42	(9.1)	1935	(11.4)	2376	(19.7)
48–49 cm	124	(26.8)	5101	(30.1)	3700	(30.6)
50–51 cm ≥52 cm	191 106	(41.3) (22.9)	6384 3513	(37.7) (20.7)	4220 1785	(34.9) (14.8)
Breastfeeding at birth			(<0.001) ^3^		(<0.001) ^4^	
Yes	431	(93.9)	12,901	(77.0)	7884	(66.7)
No	28	(6.1)	3358	(23.0)	3936	(33.3)

^1^ Birth certificates provided 8 days after birth covering Isère *département*, where Grenoble is located. The population was restricted to women (1) who gave birth in one of the 4 maternity wards of Grenoble area. (2) who were older than 18 years old when they give birth (3) whose date of last menstrual period was between March 2014 (to match with SEPAGES population) and February 2017 (no data were available after that date). ^2^ Source: 2016 French National Perinatal Survey [62]. ^3^ P-value; chi-square test (or Fisher exact test when needed) comparing the characteristics of children included in SEPAGES and children living in Grenoble area. ^4^ P-value; chi-square test (or Fisher exact test when needed) comparing the characteristics of children included in SEPAGES and children living in France (2016 French National Perinatal Survey [62]).

**Table 3 ijerph-16-03888-t003:** Biological samples collected in SEPAGES volunteers.

	Before Delivery		After Delivery					
**Matrix**	Mother	Father	Delivery (mother)	Birth (child)	2 months (child)	12 months (child)	24 months (child)	36 months (child)
**Whole blood**	1 EDTA tube (3 mL)	1 EDTA tube (3 mL)	1 EDTA tube (3 mL)	One drop				
**Serum**	5 (500 µL) aliquots	5 (500 µL) aliquots	5 (500 µL) aliquots			4 (250 µL) aliquots		4 (250 µL) aliquots
**Plasma (EDTA)**	3 (500 µL) aliquots	3 (500 µL) aliquots	3 (500 µL) aliquots			4 (250 µL) aliquots		4 (250 µL) aliquots
**Plasma Heparine**	3 (500 µL) aliquots	3 (500 µL) aliquots	3 (500 µL) aliquots					
**Buffy Coat**	1 aliquot	1 aliquot	1 aliquot			1 aliquot		1 aliquot
**Blood-RNA**	1 Tempus™ tube (3 mL)	1 Tempus™ tube (3 mL)	1 Tempus™ tube (3 mL)			1 Tempus™ tube (3 mL)		1 Tempus™ tube (3 mL)
**Placental-RNA**			3 aliquots					
**Placenta (PFA)**			1 aliquot					
**Placenta**			2 aliquots					
**Hair**	~200 mg	~200 mg	~200 mg	One strand	One strand	One strand		One strand
**Urine**	44 to 64 samples	1 spot sample	1 spot sample	1 spot sample	9 samples	9 samples		14 samples
**Stool**				3 aliquots (meconium)	3 aliquots	3 aliquots	3 aliquots	3 aliquots
**Milk**			3 (1.5 mL) aliquots					
**Buccal cells**						2 samples		2 samples
**Nasal cells**								2 samples
**Nails**								10 pieces

EDTA: Ethylenediamine tetraacetic acid (anticoagulant agent). PFA: Paraformaldehyde.

**Table 4 ijerph-16-03888-t004:** Assessment of exposure to environmental factors in the cohort participants.

Exposure	Tools/Biological Matrix	Time Points (M: Mother C: Child)
**Urban exposome—personal measures**		
PM_2.5_ concentration and oxidative potential	MicroPem (*RTI International*) ^1^	M: Around 18 and 34 gestational weeks ^2^ C: Around 2 and 36 months ^3^
Soot (weekly measurement)	MicroAeth (*AethLabs*) ^1^	M: Around 18 gestational weeks ^2^
NO_2_ mass concentration	Passive Sampler (*Passam A.G*)	M: Around 18 and 34 gestational weeks ^2^ C: Around 2, 12 and 36 months ^3^
Benzene, toluene, ethylbenzene, xylenes	Passive Sampler (*Passam A.G*)	M: Around 18 and 34 gestational weeks C: Around 2 months
Physical activity	ActiGraph accelerometer (*ActigraphCorp*) ^1^	M: Around 18 and 34 gestational weeks C: Around 2, 12 and 36 months
Noise	App NoiseTube (*NoiseTube*) ^1^	M: Around 18 and 34 gestational weeks ^2^ C: Around 2 and 36 months ^3^
Time-space activity	Dispersion model (10 m grid) of PM_2.5_, PM_10_ and NO_2_ coupled with GPS and diaries data	M: Around 18 and 34 gestational weeks ^2^ C: Around 2 and 36 months ^3^
Temperature	Thermometer DL 101T (*VoltCraft*)	M: Around 18 and 34 gestational weeks ^2^ C: Around 2 and 36 months ^3^
Cleaning and cosmetic products	Camera on a smartphone and Cobanet ^4^ smartphone application (*EpiConcept*)	M: Around 18 and 34 gestational weeks ^2^ C: Around 2 and 36 months ^3^
Drugs	Photographs and questionnaires	M: From conception onwards
**Urban exposure—estimates at the home address**		
PM_2.5_, PM_10_ and NO_2_	Dispersion model (10 m grid) of PM_2.5_, PM_10_ and NO_2_	Home addresses estimate available for whole follow-up
Temperature, atmospheric pressure	Meteorological stations and models	
**Chemical exposome** ^5^		
**Phenols** Bisphenols A, AF, B, S, F; triclosan; triclocarban; methyl, ethyl, butyl, propyl parabens; benzophenone 3.	Urine (mother child)	M: Around 18 and 34 gestational weeks ^2^ C: Around 2, 12 and 36 months ^3^
**Phthalates** MEP (DEP metabolite) MiBP (DiBP metabolite) MnBP (DBP metabolite) MBzP (BBP) MEHP (DEHP metabolite) MEHHP (DEHP metabolite) MEOHP (DEHP metabolite) MECPP (DEHP metabolite) MMCHP (DEHP metabolite) oh-MiNP (DINP metabolite) oxo-MiNP (DINP metabolite) cx-MiNP (DINP metabolite)	Urine (mother child)	M: Around 18 and 34 gestational weeks ^2^ C: Around 2, 12 and 36 months ^3^
**DINCH metabolites**	Urine (mother, child)	M: Around 18 and 34 gestational weeks ^2^
oh-MINCH oxo-MINCH oh-MPHP		C: Around 2, 12 and 36 months ^3^
**Organophosphate pesticides metabolites** DMP, DMTP, DEP, DETP, ΣDAP ^6^	Urine (mother)	Around 18 and 34 gestational weeks ^2^
**Perfluorinated compounds** 5 PFSAs (including PFOS, PFHxS), 11 PFCAs (including PFOA, PFNA), 3 FOSAs	Serum	M: Around 18 gestational weeks

PM_10_: PM with an aerodynamical diameter below 10 µm. ^1^ 405 women had at least one follow-up week during pregnancy with a MicroPem. The following devices were also used to estimate exposure to PM_2.5_ for a subsample of women: AM510 (TSI), PDR150 (Fisher), BGI (Mesa Labs). The MicroAeth (AethLab), the Actigraph (ActigraphCorp) and the application NoiseTube were used for a subsample of women. ^2^ For a subsample of pregnant women, three weeks of measurement were performed (around 18, 26 and 34 gestational weeks). ^3^ For a subsample of children, three to four weeks of measurement were performed (around 2, 9, 12 and 36 months). ^4^ The smartphone application Cobanet was used only for cleaning products. ^5^ Additional components of the chemical exposome will be assessed as part of ATHLETE EU (H2020) exposome project. ^6^ Sum of dialkylphosphate metabolites.

**Table 5 ijerph-16-03888-t005:** Health outcomes assessed in parents, foetuses and children in SEPAGES mother-child cohort.

Health Outcome	Assessment	Whom	Time Point
Foetal Growth	Ultrasound records Measurements (birth)	Foetus Newborn	12, 22, 32 gestational weeks Birth
Postnatal growth	Clinical assessments (weight, height, skin folds)	Child	At birth, 2, 12 and 36 months
	Questionnaires	Child	Every 3 to 12 months
Respiratory health	Lung function test: spirometry, exhaled NO	Mother and father	1 year after delivery (M); Inclusion (F)
	Lung function test: multiple breath washout test, tidal breathing flow-volume loops Lung function test: forced oscillation technique (FOT)	Child	2 months 36 months
	Questionnaires (respiratory symptoms and diseases)	Mother Father Child	First and third trimesters Inclusion Every 3 to 12 months
Allergy	Skin prick tests (12 allergens for mother and father; 5 allergens for the child)	Mother Father Child	1 year after delivery Inclusion 36 months
Neuro-Development/Neurological outcomes	ADBB scale WPPSI-IV Eye tracking N-Back WAIS	Child Mother	12 months 36 months 5, 12 and 24 months ^1^ 3 years after delivery
	Questionnaires VineLand MChat MacArthur SRS and BRIEF-P	Child	12 and 24 months 24 months 12 and 24 months 36 months
Cardiovascular health	Electrocardiogram (ECG), blood pressure	Mother and Father	During pregnancy and 1 year after delivery (M) At inclusion (F)
	Blood pressure	Child	At 2, 12 and 36 months

F: Father, M: Mother, NO: Nitrogen Oxide. ^1^ The eye tracking measurements were performed for a subsample of children.

**Table 6 ijerph-16-03888-t006:** Overview of the differences in design between several generations of parents-child cohorts for environmental health. This represents a schematic view aiming at making evolutions more visible.

Cohort Generation and Period	Recruitment Period and Participants	Biospecimens	Personal Measurements	Typical Example	Limitations
**First generation** (1990s and before) “**Birth cohorts**”	Birth or later. Children only generally.	After delivery only: maternal and possibly child spot biospecimens	None (questionnaire or model-based assessment exposure)	ALSPAC [87], GINI cohorts	Limited ability to investigate the effect of pregnancy exposures (besides atmospheric pollutants)
**Second generation** (2000-)	Pregnancy. Mother and child.	Pregnancy maternal spot (urine and blood) samples. Possibly DNA (mother, child, placenta) and child postnatal blood	Possibly use of a dosimeter during a single follow-up period	EDEN [88], INMA [89] cohorts	Limitations in terms of assessment of exposure to non-persistent compounds
**Third generation** (From after 2015)	Early pregnancy or preconception. Mother, father, child.	Repeated pregnancy maternal and child (possibly pooled) urine samples. Blood (mother, father, chord, offspring), DNA, RNA, possibly live cells (mother, father, offspring), placental sample, microbiome…	Repeated use of personal monitors for air pollutants, radiation, noise, temperature…. Detailed time space activity information.	SEPAGES cohort	Possible challenges to implement for a large sample size (1000-100,000 families), unless very large funding available

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
