# Peer review of "Deciphering the Impact of Early-Life Exposures to Highly Variable Environmental Factors on Foetal and Child Health: Design of SEPAGES Couple-Child Cohort"

_ijerph, 2019, doi:10.3390/ijerph16203888_

Round 1

Reviewer 1 Report

The authors satisfactorily revised their manuscript for one of the points I raised during 2nd round of review (#2). With regard to the rest of the points, #1 and #3, the authors and I agreed to leave the decision to the editor.

I have nothing to comment further.

Author Response

Following the comment from the reviewer and the Editor's recommendation, we have now deleted the presentation of the protocol of the rabbit experiment (with the accompanying table and figure) from the manuscript. We just mention in the introduction and the conclusion that an experiment has been set in parallel to the cohort, and defer the reader to the corresponding references for further details.

Reviewer 2 Report

Thank you for your changes.

With regard to p-values, could you change those that show as 0.000 on your statistical software to <0.001 as I know all those p-values shown are not exactly 0.001 (I calculated the p-value for working during pregancy SEPAGES versus whole of France).  This is the accepted statistical notation.

Author Response

We thank the reviewer for this remark; we have now replaced the values "0.001" by "<0.001", which is indeed more correct (table 1, table 2).

Reviewer 3 Report

It’s very impressive that this SEPAGES cohort study has such a rich biobank, including over 65,000 bio specimens from about 480 couple-child triads. The project design is ambitious and the authors were trying to characterize the effect of time-varying prevalent environmental factors on continuous health and biological parameters related to respiratory health, growth and neurodevelopment, as well as to explore possible underlying biological pathways with focus on those mediated by epigenetic marks, immunologic and hormonal parameters and by gut microbiota. A toxicological experiment was also performed on a relevant animal model in parallel to the cohort. As it is mentioned in “Conclusion” part, (lines 818-823) “We believe that this approach can strengthen the level of evidence regarding the health and biological (e.g., on epigenetic marks or on the microbiota) effects of early life environmental exposures such as non-persistent endocrine disruptors of the phenols and phthalates family, air pollutants with strong temporal variations including fine particulate matter, nitrogen dioxide and benzene and other components of the exosome that could be assessed from the stored samples.” However, it’s hard to justify this conclusion solo based on the current contents. From my personal opinion, I would encourage the authors to continue publishing their findings based on the collected samples rather than publishing the design alone at current stage.

Author Response

We thank the reviewer for these positive comments.

We agree that providing detailed results is generally appealing; however, the format of this manuscript corresponds to the “protocol” format proposed by the IJERPH. As agreed with the editorial office, it is acceptable not to provide results.

Round 2

Reviewer 3 Report

I understand that the format of this manuscript corresponds to the “protocol” format proposed by the IJERPH and it is acceptable not providing results. Although this SEPAGES cohort study is very impressive, it’s hard to justify the conclusions without enough results to support. As I mentioned in my last comment, I would encourage the authors to continue publishing their findings based on the collected samples. From my personal opinion, it’s not appropriate to publish the design alone at current stage.

This manuscript is a resubmission of an earlier submission. The following is a list of the peer review reports and author responses from that submission.

Round 1

Reviewer 1 Report

I have reviewed all peer review comments and the authors’ responses. I have also reviewed the revised manuscript. I am not certain that the manuscript revisions improve the quality of the manuscript. I am appreciative that the authors removed the rabbit studies, although it is not clear why they wanted to retain limited text showing that they did these studies. The added text in lines 102-111 does not adequately explain the interrelationships between epidemiology, toxicology, and exposure science as it relates to their work. Grammar and syntax also needs to be addressed throughout the manuscript.

Author Response

I have reviewed all peer review comments and the authors’ responses. I have also reviewed the revised manuscript. I am not certain that the manuscript revisions improve the quality of the manuscript. I am appreciative that the authors removed the rabbit studies, although it is not clear why they wanted to retain limited text showing that they did these studies. The added text in lines 102-111 does not adequately explain the interrelationships between epidemiology, toxicology, and exposure science as it relates to their work. Grammar and syntax also needs to be addressed throughout the manuscript.

The reason why we retained (limited) text showing that we did the rabbit experiment is that both studies were designed, funded and to some extent conducted in parallel, with a clear aim of complementarity when it comes to the characterization of the health effects of atmospheric pollutants. We believe that this represents an original and still rather rare situation of interdisciplinarity, which had been recognized by the European Research Council when it decided to support the study. We have done our best to illustrate this complementarity in the text as well as in Figure 6. Having said this, we would understand it if the Editor decided that all mentions of the rabbit experiment had to be withdrawn from the manuscript, although this would limit its originality.

We have done our best to check grammar and syntax.

Reviewer 2 Report

I confirm my previous impressions. The protocol is well written, the human study design presents several important and innovative elements and its integration with the experimental study on rabbits (and not on rodents) allows to deepen the comprehension of molecular mechanisms underlying the impact of environmental pollutants on selected health outcomes. Multi- and interdisciplinarity are crucial to increase knowledge and mitigate bias in a very complex research field such as the assessment of the association between exposure to l toxics and effects to human health. Hence, I believe that the manuscript deserves to be published.

Author Response

I confirm my previous impressions. The protocol is well written, the human study design presents several important and innovative elements and its integration with the experimental study on rabbits (and not on rodents) allows to deepen the comprehension of molecular mechanisms underlying the impact of environmental pollutants on selected health outcomes. Multi- and interdisciplinarity are crucial to increase knowledge and mitigate bias in a very complex research field such as the assessment of the association between exposure to l toxics and effects to human health. Hence, I believe that the manuscript deserves to be published.

We thank the reviewer for this comment.

Reviewer 3 Report

I found some improvement in the revised manuscript.

All of the readers of this journal are aware of the importance of toxicological studies for the designing an epidemiologic study and inference of epidemiologic findings. Lengthy addition of the statement in Introduction section (line 100-) should be deleted.

The authors still do not adequately respond to the point I commented on how they are going to use biomarker information based on multiple samplings from each subject. How are they going to withdraw relevant exposure information for each subject to relate to health outcome when biomarker concentrations temporarily vary so much as the authors shown in Fig. 3B? Without efficient strategy, the value of multiple sampling will be significantly decreased.

Author Response

3.1 I found some improvement in the revised manuscript.

All of the readers of this journal are aware of the importance of toxicological studies for the designing an epidemiologic study and inference of epidemiologic findings. Lengthy addition of the statement in Introduction section (line 100-) should be deleted.

Reply: The text referred to by the reviewer corresponds to:

“Thus, increasingly, both toxicological and epidemiological studies are capable to characterize not only the occurrence of adverse effects possibly induced by exposures, but also to point to the underlying mechanisms, which used to be a feature of toxicology alone. In spite of this increasing similarity in aims, toxicological and epidemiological studies are generally designed independently. This independent design tends to limit the overlap between these two approaches in terms of outcomes considered and, in general, limits comparability.”

Although we agree that readers familiar to both epidemiology and toxicology are most probably aware of this, not all readers are knowledgeable of both disciplines. Moreover, this explanation had been added to justify the complementarity of the epidemiological and the toxicological study, following a previous comment from another reviewer. Considering that the reviewer does not seem to consider that we state something wrong here, we would overall prefer to maintain this text, although we of course leave the final decision to the Editor.

3.2 The authors still do not adequately respond to the point I commented on how they are going to use biomarker information based on multiple samplings from each subject. How are they going to withdraw relevant exposure information for each subject to relate to health outcome when biomarker concentrations temporarily vary so much as the authors shown in Fig. 3B? Without efficient strategy, the value of multiple sampling will be significantly decreased.

Reply: Figure 3B corresponds to the biomarkers levels assessed in spot urine samples collected over a week in a pregnant woman. This (that is, the estimate from a single of these random samples) corresponds to what is commonly used in most epidemiological studies of non-persistent chemicals’ effects published so far. The consequence of relying on a spot biospecimen to assess exposures in etiological studies indeed entails strong bias, as has been previously demonstrated (e.g., Perrier et al, Epidemiology, 2016). The limited value and the strong bias entailed by these highly variable exposure estimates is one of the main motivations of our study. Our study will rely on weekly urine pools – that is, the biomarker’s level assessed from the weekly pool in each woman (or child) in the study. As we have demonstrated (Perrier et al, Epidemiology, 2016; Vernet et al, Epidemiology, 2018), using this value issued from the pooled sample in regression models relating it to the considered health outcome allows to obtain an estimate of the dose-response function that has very little bias. There is no further transformation of the exposure estimate that is required, although alternative approaches are possible if each of the (43) biospecimens collected in each woman were assessed, which would strongly increase the total assay cost and has not been done in the study. All these references are given in the manuscript, and this approach has already been used on our samples in a recent study (Warembourg et al, Int J Hyg Env Health, 2019).

We have now expanded the part explaining the statistical analysis, to make it clear that the planned analysis consists in relating the single estimate from the pooled sample to the outcome, and not exposure estimates assessed separately in each collected biospecimen:

« If women collect an identical number of biospecimens each, statistical analyses relating exposures assessed by within-subject pooling and biological parameters can be done by regression modeling like in the case of exposures assessed from spot biospecimens [37], or even in a simpler way, as correction for biospecimens’ sampling condition [82] may not be required. That is, the estimate from each (e.g., weekly) within-subject urine pool can be separately related to the outcome of interest in a (e.g., linear) regression model, adjusting for the relevant factors. In the case of between-subject variations in the number of biospecimens collected (which were limited between SEPAGES women), Perrier et al. suggested to apply weights to the regression models, whose value depend on the number of biospecimens collected in each subject [37].”

Reviewer 4 Report

Many of the tests in Table 1 should be Fisher's Exact test not chi squared as numbers in some cells are <5

Actual p-values in Table 1 would be more informative than stars

You do not say what you are going to do with the data once you have it (statistical analysis section).

It is not easy to see how the rabbit study fits with the human study

Author Response

4.1 Many of the tests in Table 1 should be Fisher's Exact test not chi squared as numbers in some cells are <5

Reply: We have now performed Fisher’s exact tests for cells with low expected p-values.

4.2 Actual p-values in Table 1 would be more informative than stars

Reply: We now indicate p-values, both in Table 1 and Table 2.

4.3 You do not say what you are going to do with the data once you have it (statistical analysis section).

Reply: We have now added a section on the planned statistical analyses (4.2, l.751 and following).

It mentions in particular:

“The overall aim of the cohort is to provide an improved characterization of the effect of time-varying prevalent environmental factors on continuous health and biological parameters related to respiratory health, growth and neurodevelopment, as well as to contribute to the unraveling of the underlying pathways possibly implied. The first analyses planned include: (1) the description of lung function trajectories in early-life; (2) the study of influences of air pollutants (PM2.5, NO2, benzene) on fetal growth, lung function and placental epigenetic marks; (3) the study of possible effects of exposure to endocrine disruptors on the same outcomes as well as on child’s neurodevelopment and gut microbiota. Projects related to the effects of temperature on health and epigenetic marks, as well as of the epigenome as a whole are also envisioned.”

4.4 It is not easy to see how the rabbit study fits with the human study.

Reply: As previously indicated, we have planned and funded both studies together so that both studies can both inform about the effects of early-life exposure atmospheric pollutants on development and health. The similarity in design between both studies is in particular shown in Figure 6.

Round 2

Reviewer 3 Report

With regard to the lengthy explanation on the relevance of animal experiment, I still do not agree with the authors and still believe that they should delete the description. But I garee with the authors in that this is left for the editor's decision.

The authors mention that they are going to prepare a "weekly pooled urine samples" to relate to health outcomes. Is this pooled urine sample a mixture of a fixed volume taken from each urine void over the week? Or, is this a mixture of all of the urine samples of the week?

The authors still include animal experiment part in the revised manuscript though it was considerably criticized. I do not accept to includ the animal experiment part and insist to delete this part from the manuscript for the reasons twice specifeid earlier. I also leave this to the editor's decision. 

Reviewer 4 Report

Thank you for adding exact p-values to the Tables 1 and 2.  For those that are highly significant, please use the accepted notation of <0.001 (if your actual p-value is below this).

Thank you for attempting a statistical analysis section.  However, what is written does not really say how you are planning on analysing the data, more the research questions explained/ reiterated and not how these are going to be answered.

The text needs further checking for grammar.